# Language Generation in the Limit: Complexity Barriers and Implications for Learning

**Marcelo Arenas** [* 1 2]  **Pablo Barceló** [* 2 3 4]  **Luis Cofré** [* 5]  **Alexander Kozachinskiy** [4]

## Abstract

Kleinberg and Mullainathan showed that language generation in the limit is always possible at the level of computability: given enough positive examples, a learner can eventually generate data indistinguishable from a target language. However, such existence results do not address feasibility. We study the sample complexity of language generation in the limit for several canonical classes of formal languages. Our results show that infeasibility already appears for context-free and regular languages, and persists even for strict subclasses such as locally threshold testable languages, as well as for incomparable classes such as non-erasing pattern languages, a well-studied class in the theory of language identification. Overall, our results establish a clear gap between the theoretical possibility of language generation in the limit and its computational feasibility.

## 1. Introduction

**Context.** The study of language identification in the limit originates in the seminal work of Gold (1967). Informally, a collection $\mathcal{L}$ of languages is said to be *identifiable in the limit* if there exists a learning algorithm that, given an infinite stream of positive examples from some target language $L \in \mathcal{L}$, produces a sequence of hypotheses that eventually stabilizes on a correct description of $L$. Gold showed that this notion is highly restrictive: even the class of regular languages is not identifiable in the limit from positive

[1]Department of Computer Science, Pontifical University of Chile [2]Instituto Milenio Fundamentos de los Datos [3]Institute for Mathematical and Computation Engineering, Pontifical Catholic University of Chile [4]National Center for Artificial Intelligence of Chile (CENIA) [5]Faculty of Mathematics, Pontifical Catholic University of Chile. Correspondence to: Marcelo Arenas <marenas@uc.cl>, Pablo Barceló <pbarcelo@uc.cl>, Luis Cofré <luis.cofr@uc.cl>, Alexander Kozachinskiy <alexander.kozachinskyi@cenia.cl>.

*Proceedings of the 43rd International Conference on Machine Learning*, Seoul, South Korea. PMLR 306, 2026. Copyright 2026 by the author(s).

data alone, whereas finite families trivially are. Angluin subsequently refined this negative picture by providing a structural characterization of families that are identifiable in the limit (Angluin, 1980a). Despite these refinements, few natural language families satisfy these conditions. As a result, Gold–Angluin style unlearnability results have often been interpreted as formal support for the *poverty of the stimulus* hypothesis in linguistics, suggesting that purely data-driven learning is insufficient without strong innate biases or prior structure (Pinker, 2013; Pearl, 2021).

A notable exception to this pessimistic landscape is provided by *non-erasing pattern languages* (Angluin, 1980b). These languages are generated by words over constants and variables, where each variable must be instantiated by a nonempty word. Pattern languages are expressive enough to capture nontrivial structural regularities, yet sufficiently constrained to form a canonical example of a nontrivial family that is learnable in Gold's model (Angluin, 1980a).

The recent empirical success of large language models (LLMs) contrasts with these classical learnability limitations. Although trained on finite samples and exposed primarily to positive data, LLMs appear to internalize a wide range of syntactic and semantic regularities, suggesting that useful generalization may be possible without explicitly identifying an underlying grammar. Motivated by this tension, Kleinberg and Mullainathan introduced the notion of *language generation in the limit* (Kleinberg & Mullainathan, 2024), which shifts the focus from identifying a language to generating strings of a target language, unseen in the training data. Unlike identification, language generation in the limit is always possible at the level of countable families of languages. This establishes a fundamental asymmetry between learning and generation and motivates closer study of generation as a lens for understanding generalization.

**Problem.** While language generation in the limit is computable in full generality, this fact alone says little about *feasibility*. From a computational standpoint, the key question is how many examples are required before a generator can reliably produce strings indistinguishable from those of a target language. If this sample complexity grows too rapidly with the size or structure of the language family, then

generation in the limit remains a purely existential result.

**Our results.** We study the sample complexity of language generation in the limit for finite families of languages drawn from several canonical language classes. While non-computable bounds are easy to obtain for highly pathological families, we show that severe feasibility barriers already arise for central and well-studied classes.

- *Context-free languages:* We show that there is no computable bound on the number of examples required to guarantee successful generation in the limit, even for families specified by pushdown automata or context-free grammars. Thus, non-computability already arises for a canonical and extensively studied class.

- *Regular languages:* Restricting attention to regular languages restores computability, but at a high cost. For finite families of regular languages represented by (deterministic) finite automata or regular expressions, any generator requires a double-exponential number of examples in the size of the family.

- *LTT languages:* We next consider *locally threshold testable* (LTT) languages, a robust and well-studied subclass of the regular languages. Informally, LTT languages are determined by local counting conditions on short substrings and admit equivalent combinatorial, logical, and algebraic characterizations. For this class, the sample complexity of generation in the limit is precisely single exponential. While this improves over the double-exponential bound for arbitrary regular languages, the resulting complexity remains infeasible.

- *Patterns:* Finally, we study the previously mentioned non-erasing pattern languages, which are incomparable with the regular and the LTT languages. Despite their favorable properties for identification in the limit, we establish an exponential lower bound on the sample complexity of generation over finite families.

## 2. Language Generation

**Generation in the limit.** Let $\Sigma$ be a finite alphabet and let $\Sigma^*$ denote the set of all finite words over $\Sigma$. A *language* over $\Sigma$ is a subset of $\Sigma^*$. A *generator* is a function $G$ mapping a finite set of words from $\Sigma^*$ to a word in $\Sigma^*$.

**Definition 2.1** (Language generation in the limit). A set $\mathcal{F}$ of infinite languages over $\Sigma$ is *generatable in the limit* if there is a generator $G$ such that, for every $L \in \mathcal{F}$ and ordering $w_1, w_2, \ldots$ of the words in $L$, there is $m \geq 1$ with $G(\{w_1, \ldots, w_n\}) \in L \setminus \{w_1, \ldots, w_n\}$ for all $n \geq m$. $\square$

Language generation in the limit was introduced by Kleinberg & Mullainathan (2024), who showed that every count-

able family of languages is generatable in the limit. However, this notion is inherently non-quantitative: the point of convergence may depend arbitrarily on the target language and on the ordering of its examples, and therefore does not yield a meaningful notion of sample complexity.

**Uniform generation.** To reason about feasibility, one needs guarantees that hold after a fixed number of examples, uniformly across the family. This motivates the notion of *uniform generation*, also introduced in (Kleinberg & Mullainathan, 2024). In particular, they observe that every finite family of languages admits such a uniform bound, making uniform generation the appropriate setting for studying generation from finite collections of languages.

**Definition 2.2** (Uniform generation). Let $m \geq 1$. A set $\mathcal{F}$ of infinite languages is $m$-*generatable* if there exists a generator $G$ such that, for every $L \in \mathcal{F}$ and every finite $X \subseteq L$ with $|X| \geq m$, it holds that $G(X) \in L \setminus X$.

Such a generator $G$ is called an $m$-*generator* for $\mathcal{F}$. $\square$

Families that are $m$-generatable for some $m \in \mathbb{N}$ were characterized by Raman et al. (2025). We recall a streamlined version of this characterization, adapted to our setting.

**Proposition 2.3.** *A set $\mathcal{F}$ of infinite languages is not $m$-generatable if and only if there exists a non-empty $\mathcal{S} \subseteq \mathcal{F}$ such that $\bigcap_{L \in \mathcal{S}} L$ is finite and has size at least $m$.*

*Proof.* Suppose there exists a non-empty $\mathcal{S} \subseteq \mathcal{F}$ such that $X = \bigcap_{L \in \mathcal{S}} L$ is finite and $|X| \geq m$. Assume for contradiction that an $m$-generator $G$ exists. If $G(X) \in X$, then for any $L \in \mathcal{S}$ we have $G(X) \in L$, contradicting $G(X) \in L \setminus X$. If $G(X) \notin X$, then $G(X) \notin L$ for some $L \in \mathcal{S}$, again a contradiction.

Conversely, assume no such $\mathcal{S}$ exists. For a finite set $X \subseteq \Sigma^*$, define $\mathcal{S}_X = \{L \in \mathcal{F} \mid X \subseteq L\}$ and $A_X = \bigcap_{L \in \mathcal{S}_X} L$. Define $G(X)$ to be any element of $A_X \setminus X$ when $\mathcal{S}_X$ is non-empty. If $X \subseteq L' \in \mathcal{F}$ with $|X| \geq m$, then $A_X \subseteq L'$ has size at least $m$ and, by assumption, must be infinite; hence $A_X \setminus X \neq \emptyset$ and $G(X) \in L' \setminus X$. $\square$

When $\mathcal{F}$ is finite, it is necessarily $m$-generatable for some $m$ (take $m$ larger than the size of any finite intersection of the form $\bigcap_{L \in \mathcal{S}} L$, there are finitely many such intersections for a finite $\mathcal{F}$). However, this observation concerns existence only. To study feasibility, one must quantify how large $m$ must be as a function of the size of $\mathcal{F}$ and the descriptive complexity of its languages. Accordingly, in the remainder of the paper we focus on uniform generation for finite families and analyze the sample complexity required for generation in the limit across several canonical language classes.

**Related work.** Before turning to our technical results, we briefly review related work on language generation. Raman et al. (2025) introduce the notion of *non-uniform generation*, where the number of examples required for generation may depend on the target language, but not on the order in which examples are presented. They distinguish this notion from uniform generation and generation in the limit, and relate it to classical learning frameworks such as PAC and online learning. Subsequently, Hanneke et al. (2025) study closure properties of these generation notions under language union.

A different line of work focuses on the *breadth* of generation, that is, on how representative the generated subset is of the target language. Kalavasis et al. (2025) show that one cannot, in general, converge to generating all unseen elements of the target language, even under a statistical (rather than adversarial) model of the example presentation. On the positive side, Kleinberg & Wei (2025) demonstrate that it is possible to generate a subset of the target language with positive density. Finally, Peale et al. (2025) propose a notion of *representative* language generation inspired by algorithmic fairness, requiring that the relative proportions of different segments of the language in the generated output approximately match those observed in the data.

These works address complementary aspects of language generation, such as robustness, coverage, and representativeness. In contrast, our focus is on the *sample complexity* of uniform generation, and on understanding how it depends on the structural properties of the underlying language class.

# 3. Results for Context-Free Languages

We begin with context-free languages, a canonical and extensively studied class in formal language theory. Our objective is to understand how many examples are required to generate from an unknown context-free language drawn from a finite family. We show that, even in this restricted setting, there is no computable bound on the number of examples sufficient for uniform generation, already for families consisting of only two infinite languages.

Throughout this section, we assume familiarity with Turing machines and the undecidability of the halting problem.

## 3.1. Context-free grammars

Context-free grammars are a standard and widely used formalism for specifying context-free languages. We briefly recall their definition A *context-free grammar* (CFG) is a tuple $G = (\mathcal{X}, \Sigma, R, S)$, where:

- $\mathcal{X}$ is a finite set of *variables*, denoted by $X, Y, \ldots$.

- $\Sigma$ is a finite alphabet of *constants*, disjoint from $\mathcal{X}$ and denoted by $a, b, \ldots$.

- $R \subseteq \mathcal{X} \times (\mathcal{X} \cup \Sigma)^*$ is a finite set of *production rules*.

- $S \in \mathcal{X}$ is the *start* variable.

We define the *one-step derivation relation* $\Rightarrow_G \subseteq (\mathcal{X} \cup \Sigma)^* \times (\mathcal{X} \cup \Sigma)^*$ as follows. For words $u, v \in (\mathcal{X} \cup \Sigma)^*$, we write $u \Rightarrow_G v$ if there exist words $x, y \in (\mathcal{X} \cup \Sigma)^*$, a variable $X \in \mathcal{X}$, and a rule $(X \to \alpha) \in R$ such that

$$u = xXy \quad \text{and} \quad v = x\alpha y.$$

The relation $\Rightarrow_G^*$ denotes the reflexive and transitive closure of $\Rightarrow_G$. A word $w \in (\mathcal{X} \cup \Sigma)^*$ is said to be *derived* from a variable $X \in \mathcal{X}$ if $X \Rightarrow_G^* w$.

The language generated by $G$ is

$$L(G) = \{ w \in \Sigma^* \mid S \Rightarrow_G^* w \}.$$

A language is *context-free* if it is generated by some context-free grammar.

**Example 3.1.** Let $\Sigma = \{a, b\}$. The language $\{a^n b^n \mid n \geq 0\}$ is context-free. It is generated by the grammar $G = (\{S\}, \{a, b\}, R, S)$ with production rule $S \to aSb \mid \epsilon$, where $\epsilon$ is the empty word. In contrast, the language $\{a^n b^n c^n \mid n \geq 0\}$ is not context-free. $\square$

## 3.2. Non-computable sample complexity

We now state our main result for context-free languages.

**Theorem 3.1.** *There is no algorithm that, given two infinite CFGs $G_1$ and $G_2$, outputs a number $m \in \mathbb{N}$ such that the family $\{L(G_1), L(G_2)\}$ is $m$-generatable. The algorithm may diverge if either $L(G_1)$ or $L(G_2)$ is finite.*

*Proof.* By reduction from the halting problem.

**Lemma 3.2.** *There exists an algorithm that, given a Turing machine $M$ with empty input, constructs two CFGs $G_1$ and $G_2$ such that, if $M$ halts in exactly $t$ computation steps, then*

$$|L(G_1) \cap L(G_2)| = 2^t.$$

Assuming Lemma 3.2, we show that the existence of the algorithm postulated in Theorem 3.1 would imply decidability of the halting problem. Given a Turing machine $M$, we construct CFGs $G_1$ and $G_2$ as in Lemma 3.2. It is a classical result that finiteness of context-free languages is decidable: given a context-free grammar $G$, one can determine whether $L(G)$ is finite and, if so, compute its cardinality (see, e.g., (Ginsburg, 1966)).

We distinguish three cases:

1. If $L(G_1)$ is finite, choose $t$ such that $2^t > |L(G_1)| \geq |L(G_1) \cap L(G_2)|$. Then $M$ cannot halt in $t$ or more steps. We simulate $M$ for $t - 1$ steps and conclude that it does not halt if no halting configuration is reached.

2. The case where $L(G_2)$ is finite is symmetric.

3. If both $L(G_1)$ and $L(G_2)$ are infinite, apply the algorithm from Theorem 3.1 to obtain $m$ such that $\{L(G_1), L(G_2)\}$ is $m$-generatable. Choose $t$ with $m \leq 2^t$. By Proposition 2.3, if $M$ were to halt in $s \geq t$ steps, then $|L(G_1) \cap L(G_2)| = 2^s \geq 2^t \geq m$, contradicting $m$-generatability. Hence $M$ cannot halt in $t$ steps, and it suffices to simulate $M$ for $t - 1$ steps.

Therefore, the assumed algorithm cannot exist. $\square$

*Proof of Lemma 3.2.* Assume that $M$ halts in exactly $t$ computation steps, and let $C_1, \ldots, C_t$ be the sequence of configurations of $M$ (encoded as words over a finite alphabet in the standard way). We add two fresh delimiter constants $\#_0$ and $\#_1$. We construct CFGs $G_1$ and $G_2$ such that $L(G_1) \cap L(G_2)$ consists precisely of the words

$$C_1 \#_{i_1} C_2^R \#_{i_2} \cdots C_t^R \#_{i_t}$$

if $t$ is even, and

$$C_1 \#_{i_1} C_2^R \#_{i_2} \cdots C_t \#_{i_t}$$

if $t$ is odd, where each $i_j \in \{0, 1\}$ and $w^R$ denotes the reversal of $w$.

The CFGs enforce the following conditions:

- $C_1$ encodes the initial configuration of $M$;

- $C_t$ encodes the unique accepting configuration;

- for each $i < t$, the pair $(C_i, C_{i+1})$ represents a valid transition of $M$.

The verification of transition correctness is distributed between $G_1$ and $G_2$, alternating between odd and even indices. Each grammar checks only local consistency conditions between consecutive configurations, which can be enforced by context-free productions using standard techniques.

Each choice of the delimiter sequence $(i_1, \ldots, i_t)$ yields a distinct word in $L(G_1) \cap L(G_2)$, and all such words arise in this way. Hence, $|L(G_1) \cap L(G_2)| = 2^t$. $\square$

### 3.3. Other descriptions for context-free languages

The lower bound established in Theorem 3.1 is robust with respect to the choice of formalism used to specify context-free languages. In particular, the same non-computable sample complexity arises when languages are given by *pushdown automata* (PDAs), that is, finite-state machines equipped with an unbounded stack, since every CFG can be translated into an equivalent PDA, and vice versa, with only polynomial overhead (see, e.g., (Hopcroft et al., 2007)).

Moreover, the construction underlying the proof can be carried out using *deterministic* PDAs. Since deterministic PDAs form a strict subclass of the context-free languages, this shows that the source of non-computability is not nondeterminism or excessive expressive power, but already manifests under strong determinism constraints.

## 4. Results for Regular Languages

We now consider $m$-generatability for finite families of regular languages specified by regular expressions or finite automata. We prove that, in the worst case, the sample complexity of uniform generation is double exponential in the size of the family.

### 4.1. Regular expressions and finite automata

Regular expressions are a standard and widely used formalism for specifying regular languages. We briefly recall their definition. Let $\Sigma$ be a finite alphabet. The set of *regular expressions* over $\Sigma$ is defined by the grammar

$$r ::= \emptyset \mid \varepsilon \mid a \mid (r + r) \mid (rr) \mid r^*,$$

where $a$ ranges over $\Sigma$. Each regular expression $r$ defines a language $L(r) \subseteq \Sigma^*$, where $\emptyset$ defines $\emptyset$, $\varepsilon$ defines $\{\varepsilon\}$, each $a \in \Sigma$ defines $\{a\}$, $(r + s)$ defines $L(r) \cup L(s)$, $(rs)$ defines $\{uv \mid u \in L(r), v \in L(s)\}$, and $r^*$ defines $\bigcup_{k \geq 0} L(r^k)$, with $r^0 = \varepsilon$ and $r^k = (rr^{k-1})$ for $k \geq 1$.

A language over $\Sigma$ is *regular* if it is defined by some regular expression over $\Sigma$.

**Example 4.1.** The regular language defined by $a^*b^*$ consists of all words over $\{a, b\}$ in which no occurrence of $a$ appears after an occurrence of $b$. In contrast, the context-free language $\{a^n b^n \mid n \geq 0\}$ is not regular. $\square$

A regular language can alternatively be specified by a finite automaton. Recall that a *non-deterministic finite automaton (NFA)* over an alphabet $\Sigma$ is a tuple $\mathcal{A} = (Q, q_0, F, \Delta)$, where (a) $Q$ is a finite set of *states*, (b) $q_0 \in Q$ is the *initial state*, (c) $F \subseteq Q$ is the set of *final* states, and (d) $\Delta \subseteq Q \times \Sigma \times Q$ is a set of *transition rules*.

The acceptance of a word $w$ by $\mathcal{A}$ is defined as follows. Assuming that $w = a_1 \ldots a_n$ with $a_i \in \Sigma$ for every $i \in \{1, \ldots, n\}$, a *run* of $\mathcal{A}$ on $w$ is a function $\rho : \{0, 1, \ldots, n\} \rightarrow Q$ such that $\rho(0) = q_0$ and $(\rho(i), a_{i+1}, \rho(i + 1)) \in \Delta$ for every $i \in \{0, \ldots, n - 1\}$. Such a run is *accepting* if $\rho(n) \in F$. The language accepted by $\mathcal{A}$, denoted by $L(\mathcal{A})$, is the set of words $w \in \Sigma^*$ such that there exists an accepting run of $\mathcal{A}$ on $w$.

An NFA $\mathcal{A} = (Q, q_0, F, \Delta)$ over an alphabet $\Sigma$ is said to be *deterministic*, or a deterministic finite automaton (DFA), if for every $q \in Q$ and $a \in \Sigma$, there exists at most one

state $q' \in Q$ such that $(q, a, q') \in \Delta$. Every NFA can be translated into a DFA with an exponential blowup that is unavoidable for some families of NFAs.

Every regular expression $r$ can be translated in polynomial time into an NFA $\mathcal{A}$ such that $L(r) = L(\mathcal{A})$; moreover, if $r$ has length $n$, then $\mathcal{A}$ has at most $n + 1$ states (Ellul et al., 2005). Conversely, while every NFA can be converted into an equivalent regular expression, this transformation may incur an unavoidable exponential blowup. Thus, regular languages admit equivalent descriptions by regular expressions, NFAs, or DFAs.

## 4.2. Double-exponential sample complexity

We now establish our main result on uniform generation for finite families of regular languages. The bounds we obtain do not depend on the chosen representation and should be viewed as intrinsic to the class of regular languages. We first prove the result for languages specified by finite automata, and then extend it to regular expressions. The latter does not follow directly, since translations from automata to regular expressions can be exponentially more succinct.

**Definition 4.1.** Let $\mathcal{F}$ be a finite family of infinite languages. Its **sample complexity**, denoted by $\mathrm{sc}(\mathcal{F})$, is the minimal $m \in \mathbb{N}$ such that $\mathcal{F}$ is $m$-generatable.

In the following theorem we study the maximal possible sample complexity of a family of $n$ regular languages, each recognizable by an $s$-state NFA (as a funcion of $s, n$).

**Theorem 4.2.** *Let $s, n, a \in \mathbb{N}$, $a \geq 2$.*

a) *Let $\mathcal{F}$ be a family of $n$ regular infinite languages over an alphabet of size $a$ such that each $L \in \mathcal{F}$ is accepted by an NFA with at most $s$ states. Then*

$$\mathrm{sc}(\mathcal{F}) \leq a^{s^n}.$$

*Moreover, the same result holds if each language in $\mathcal{F}$ is defined by a regular expression of length at most $s - 1$.*

b) *Assume that $a = 2$ and $n$ is a power of 2. Then there exists a family of $n$ regular infinite languages over the binary alphabet such that each $L \in \mathcal{F}$ is accepted by an NFA with at most $s$ states, and*

$$\mathrm{sc}(\mathcal{F}) > 2^{(\Omega(s/\log n))^n}.$$

*Moreover, each $L \in \mathcal{F}$ is, in fact, accepted by a DFA with at most $s$ states, and is defined by a regular expression of length $O(s^2)$.*

Note that the lower bound in item b) trivializes for families of size $|\mathcal{F}| = 2^{\Omega(s)}$. However, it is known that there are

$s^{\Omega(s)}$ distinct regular languages, recognizable by $s$-state DFAs (Domaratzki, 2006). It is an interesting open question whether for families of such size one can obtain a similar double-exponential lower bound.

*Proof of Theorem 4.2.* We first establish *a)*. By the standard product construction, each intersection $\bigcap_{L \in \mathcal{S}} L$, for $\mathcal{S} \subseteq \mathcal{F}$, is recognizable by a finite automaton with at most $s^{|\mathcal{F}|}$ states. If this intersection is finite, it cannot have a word of length $s^{|\mathcal{F}|}$ or larger. Hence, this intersection has size less than $a^{s^{|\mathcal{F}|}}$. By Proposition 2.3, the family is $a^{s^{|\mathcal{F}|}}$-generatable. Moreover, we obtain that the same result holds if each language in $\mathcal{F}$ is defined by a regular expression of length at most $s - 1$, since every such regular expression can be translated into an NFA with at most $s$ states.

We now proceed to the proof of *b)*. Denote $\ell = \log_2 n$. Let $w \in \{0,1\}^\ell$. Next, let $* < w$ denote the set of all words in $\{0,1\}^\ell$ that precede $w$ in the lexicographic order. Let $m \in \mathbb{N}$ be a parameter to be specified later. Define $L_w^m$ to be the following language. First, this language will consist only of words whose length is a multiple of $\ell$. We thus will think of words in $L_w^m$ as split into blocks of length $\ell$. We put a word into $L_w^n$ if and only if the sequence of its blocks in *odd* positions satisfies the following: there is no consecutive fragment of this sequence that has more than $m$ blocks $w$ but no block from $* < w$. In turn, the blocks in even positions can be filled arbitrarily.

We show that the family $\mathcal{F} = \left\{ L_w^m \mid w \in \{0,1\}^\ell \right\}$ satisfies all the requirements for an appropriate choice of $m$. First, $|\mathcal{F}| = 2^\ell$ by construction. It is easy to see that $L_w^m$ is infinite for every $w \in \{0,1\}^\ell$ (for instance, it contains all words without $w$-blocks).

We now show $L_w^m$ is recognizable by an $O(\ell \cdot m)$-state DFA for every $w \in \{0,1\}^\ell$. This DFA counts the number of $w$-blocks, resetting it to 0 after any block from $* < w$ and checking that this number never exceeds $m$. We thus need a counter, taking $m + O(1)$ possible values. For each value of the counter, we will have an $O(\ell)$-state subautomaton to process blocks in odd positions (remembering the current value of the counter), and $O(\ell \cdot m)$ states overall. Each of these subautomata tracks for how long the current block coincides with $w$ when we read it from left to right. It suffices to have $\ell + O(1)$ states for this – one per prefix of $w$. In the end, we will be able to decide whether the current block is less than $w$ (in which case we reset the counter to 0), equal to $w$ (in which case we increase the counter) or bigger than $w$ (in which case we do nothing with the counter) in the lexicographic order.

Therefore, we can fix $m = \Theta(s/\ell)$ so that the upper bound on the number of states of the DFA from the previous paragraph does not exceed $s$. This ensures that any $L \in \mathcal{F}$ is

indeed recognizable by a DFA with at most $s$ states.

We now show that $\mathcal{F}$ is not $2^{m^{|\mathcal{F}|}}$-generatable. This implies that

$$\text{sc}(\mathcal{F}) > 2^{m^{|\mathcal{F}|}} = 2^{(\Omega(s/\log n))^n},$$

and we get all the requirements for our family (except the requirement about regular expressions). To show this, we use Proposition 2.3, showing that the intersection of languages in the family:

$$\bigcap_{w \in \{0,1\}^\ell} L_w^m \tag{1}$$

is finite but has size at least $2^{m^{|\mathcal{F}|}}$.

First, we show that (1) is finite. Let $w_0 < w_1 < \ldots < w_{2^\ell - 1}$ the words in $\{0,1\}^\ell$ going in the lexicographic order. If a word belongs to $L_{w_0}^m$, it has at most $m$ odd $w_0$-blocks. If it additionally belongs to $L_{w_1}^m$, then we have at most $m + 1$ "spaces" between $w_0$-blocks, and in each space we can put at most $m$ blocks $w_1$. Thus, in total, we can have at most $(m + 1)m$ blocks $w_1$.

Likewise, assuming that we have obtained a finite upper bound on the number of odd blocks for each word in $* < w$, we get the following upper bound on the number of odd $w$-blocks – at most $m$ times (the sum of bounds for each word in $* < w$ plus 1). Indeed, in each space between a block from $* < w$ we can put at most $m$ blocks $w$, and the number of spaces is the number of corresponding blocks plus 1. This implies that for words in (1) we have a finite upper bound on the number of occurrences of each word in odd blocks, meaning that all words in these intersections have length at most some finite number.

We now show that size of (1) is at least $2^{m^{|\mathcal{F}|}}$. It is enough to construct a word in (1) with at least $m^{|\mathcal{F}|}$ odd blocks (as we can have the same number of even blocks and fill them arbitrarily). We construct such a word as follows: we put $m$ blocks $w_0$, then in all the spaces we put $m$ blocks $w_1$, and so on. Each time, the number of blocks increases by a factor of at least $m$ – before each old block appears $m$ new ones. Hence, after doing this for all $2^\ell$ words in $\{0,1\}^\ell$, we will obtain at least $m^{2^\ell} = m^{|\mathcal{F}|}$ blocks.

To conclude the proof of the theorem, we need to define each language $L_w^m$ with a regular expression of length $O(\ell^2 \cdot m^2) = O(s^2)$. Assume that $w = a_1 \cdots a_\ell$, where $a_i \in \{0,1\}$ for every $i \in \{1, \ldots, \ell\}$. Moreover, assume that $Z$ is the set of positions $i \in \{1, \ldots, \ell\}$ such that $a_i = 0$, and $O$ is the of positions $i \in \{1, \ldots, \ell\}$ such that $a_i = 1$. Then for every $i \in O$, let $prec_i$ be a regular expression that defines the set of words $w' \in \{0,1\}^\ell$ such that, $w'$ precedes $w$ in the lexicographic order because the symbol in position $i$ of $w'$ is 0, and $w, w'$ have the same symbols in positions 1 to $i - 1$. Formally, this regular expression is defined as follows:

$$prec_i = a_1 \cdots a_{i-1} 0 \underbrace{(0+1) \cdots (0+1)}_{\ell - i \text{ times}}.$$

Then the following regular expression defines the set of words in $\{0,1\}^\ell$ that precede $w$ in the lexicographic order:

$$prec = \sum_{i \in O} prec_i.$$

In the same way, for every $i \in Z$, let $follow_i$ be a regular expression that defines the set of words $w' \in \{0,1\}^\ell$ such that, $w'$ follows $w$ in the lexicographic order because the symbol in position $i$ of $w'$ is 1, and $w, w'$ have the same symbols in positions 1 to $i - 1$. Formally, this regular expression is defined as follows:

$$follow_i = a_1 \cdots a_{i-1} 1 \underbrace{(0+1) \cdots (0+1)}_{\ell - i \text{ times}}.$$

And then the following regular expression defined the set of words in $\{0,1\}^\ell$ that follow $w$ in the lexicographic order:

$$follow = \sum_{i \in Z} follow_i.$$

Now for every $i \in \{0, \ldots, m\}$, define $block_i$ as a regular expression that defines the sequences of blocks that contain $i$ times the block $w$. Formally, assuming that $r^0 = \varepsilon$ and $r^i = (rr^{i-1})$ for every $i \geq 1$, we have that:

$$block_i = follow^*(w\,follow^*)^i.$$

With this notation, the following regular expression defines the language $L_w^m$:

$$r_w^m = \left(\left(\sum_{i=0}^{m} block_i\right) prec\right)^*.$$

To conclude the proof, we need to show that the size of $r_w^m$ is $O(\ell^2 \cdot m^2)$. To see why this is the case, we first note that the size of each $prec_i$ is $O(\ell)$, and so the size of $prec$ is $O(\ell^2)$. Analogously, we have that the size of $follow$ is $O(\ell^2)$. On the other side, the size of $block_i$ is $O(\ell^2 \cdot i)$, which is $O(\ell^2 \cdot m)$ if $i \leq m$. Hence, the size of $\sum_{i=0}^{m} block_i$ is $O(\ell^2 \cdot {}^2)$, from which we deduce that the size of $r_w^m$ is $O(\ell^2 \cdot m^2)$. This concludes the proof of the theorem. $\square$

## 5. Results for LTT Languages

In the previous section, we showed that uniform generation for regular languages can require a double-exponential number of examples, both when languages are represented by regular expressions and by finite automata. This naturally raises the question of whether imposing additional structure on regular languages can reduce the sample complexity

of generation. In this section, we address this question by focusing on *locally threshold testable* (LTT) languages, a robust and expressive subclass of the regular languages admitting several equivalent characterizations (see, e.g., (Place et al., 2014)). We show that restricting attention to LTT languages indeed leads to a strict reduction in complexity: the sample complexity of uniform generation becomes single exponential. While this represents a substantial improvement over the double-exponential bound for general regular languages, the resulting complexity remains prohibitively large, showing that even strong structural restrictions are insufficient to make generation feasible.

## 5.1. LTT languages

Given an alphabet $\Sigma$, a word $w' \in \Sigma^*$ is a *subword* of a word $w \in \Sigma^*$ if there exist $u, v \in \Sigma^*$ such that $w = uw'v$. If $u = \varepsilon$, then $w'$ is a *prefix* of $w$, and if $v = \varepsilon$, then $w'$ is a *suffix* of $w$. A *profile* over $\Sigma$ is a tuple $P = (pr, su, In)$, where $pr, su \in \Sigma^*$ are respectively a prefix and a suffix, and $In$ is a finite set of constraints of the form $(w, \theta t)$, with $w \in \Sigma^*$, $\theta \in \{\leq, <, =, >, \geq\}$, and $t \in \mathbb{N}$ given in unary.[1] The language defined by a profile $P$, denoted $L(P)$, consists of all words $x \in \Sigma^*$ such that $pr$ is a prefix of $x$, $su$ is a suffix of $x$, and for every $(w, \theta t) \in In$, the number of occurrences of $w$ as a subword of $x$ satisfies the constraint $\theta t$. Moreover, a language $L \subseteq \Sigma^*$ is *locally threshold testable* (LTT) if there exists a finite sequence of profiles $(P_1, \ldots, P_n)$ such that $L = \bigcup_{i=1}^{n} L(P_i)$.

**Example 5.1.** Consider the alphabet $\Sigma = \{1, \ldots, n\}$ and the profile $P = (\varepsilon, \varepsilon, In)$, where

$$In = \{(1, = 1), (2, = 1), \ldots, (n, = 1)\}.$$

Since $\varepsilon$ is both a prefix and a suffix of every word, the profile imposes no constraints on the beginning or end of a word. The constraints in $In$ instead require that each symbol $i \in \{1, \ldots, n\}$ occur exactly once. Consequently, $L(P)$ consists precisely of all permutations of the alphabet $\Sigma$, and hence is a finite language of size $n!$. $\square$

## 5.2. Single-exponential sample complexity

It is easy to see that for every profile $P$, the language $L(P)$ is accepted by an NFA: prefix and suffix constraints are local, and each counting constraint in $In$ can be implemented by a finite-state counter. Since regular languages are closed under finite union, every LTT language is regular. However, translating an LTT profile into an equivalent automaton can incur an exponential blowup. Combining this translation with the general bounds from Theorem 4.2 therefore yields a triple-exponential upper bound on the sample complexity when LTTs are given as profiles, and a double-exponential

bound when they are given as NFAs.

We show that these bounds are not inherent to LTT languages. In fact, every finite family of LTTs admits uniform generation with a *single-exponential* bound, demonstrating that the structure of LTT profiles can be exploited to obtain substantially more efficient generation.

**Theorem 5.1.** *The following statements hold:*

a) *Let $s \in \mathbb{N}$, and $\mathcal{F}$ be a finite family of infinite languages over an alphabet of size $a \geq 2$ such that each $L \in \mathcal{F}$ is defined by an LTT of size at most $s$. Then $\mathcal{F}$ is $a^{3s|\mathcal{F}|((s|\mathcal{F}|)^2+1)^2}$-generatable.*

b) *For every $n \in \mathbb{N}$ with $n \geq 4$, there exists a family $\mathcal{F}$ of two infinite languages over an alphabet of size $n$ such that each $L \in \mathcal{F}$ is defined by a profile of length at most $7(n+1)$, and $\mathcal{F}$ is not $2^n$-generatable.*

*Proof.* We first establish *b)*. The proof builds on the construction in Example 5.1. We consider a family of two LTT languages, since generation from a single infinite language is always trivial, and nontrivial lower bounds necessarily require at least two languages.

Let $\Sigma = \{1, \ldots, n\}$ with $n \geq 4$, $P_1 = (\varepsilon, \varepsilon, In_1)$ be a profile such that $In_1 = \{(i, \leq 1) : i \text{ is odd}\} \cup \{(i, \geq 1) : i \text{ is even}\}$, and $P_2 = (\varepsilon, \varepsilon, In_2)$ be a profile such that $In_2 = \{(i, \geq 1) : i \text{ is odd}\} \cup \{(i, \leq 1) : i \text{ is even}\}$. Moreover, define $\mathcal{F} = \{L(P_1), L(P_2)\}$. Both $L(P_1)$ and $L(P_2)$ are infinite because $In_1$ allows arbitrarily many occurrences of even symbols and $In_2$ allows arbitrarily many occurrences of odd symbols. On the other hand, the intersection $|L(P_1) \cap L(P_2)| = n!$ since this intersection is defined by the profile $P = (\varepsilon, \varepsilon, In)$ with $In = \{(1, = 1), (2, = 1), \ldots, (n, = 1)\}$, and we know from Example 5.1 that $|L(P)| = n!$. Hence, we deduce from Proposition 2.3 that $\mathcal{F}$ is not $n!-$generatable. As $2^n \leq n!$ for $n \geq 4$, we conclude that $\mathcal{F}$ is not $2^n-$generatable. To finish, notice that the length of both $P_1$ and $P_2$ is bounded by $7(n+1)$.

We now prove *a)*. Let $\mathcal{F} = \{L_1, \ldots, L_\ell\}$ be a finite family of infinite languages, where each $L_i$ is defined by an LTT of size at most $s$. Write each $L_i$ as a finite union of profiles $(P_1^i, \ldots, P_{m_i}^i)$. To show that $\mathcal{F}$ is $a^{3s\ell((s\ell)^2+1)^2}$-generatable, it suffices by Proposition 2.3 to prove that for every index set $I \subseteq \{1, \ldots, \ell\}$ such that $\bigcap_{i \in I} L_i$ is finite,

$$\left| \bigcap_{i \in I} L_i \right| < a^{3s\ell((s\ell)^2+1)^2}. \tag{2}$$

*Step 1: Intersections of profiles.* The intersection of two profiles of sizes $n_1$ and $n_2$ can be represented by a profile of size at most $n_1 + n_2$, or is empty. This follows by taking the longer compatible prefix and suffix (if they are compatible),

---

[1]This means that $t$ is given as a string $a^t$ for some symbol $a \in \Sigma$. For simplicity, we use the notation $t$ instead of $a^t$.

and uniting the sets of counting constraints; if prefixes or suffixes are incompatible, the intersection is empty. By iterating this construction, the intersection of $r$ profiles of size at most $n$ is represented by a profile of size at most $rn$.

*Step 2: Representing finite intersections.* By distributivity of intersection over union, the language $\bigcap_{i \in I} L_i$ is defined by an LTT consisting of $m = \prod_{i \in I} m_i$ profiles, each obtained as the intersection of one profile from each $L_i$. Each such profile has size at most $s|I| \leq s\ell$. Since $\bigcap_{i \in I} L_i$ is finite by assumption, each of these profiles defines a finite language.

*Step 3: Bounding word length.* We now use the following combinatorial fact.

**Lemma 5.2.** *If $P$ is a profile of size $r$ such that $L(P)$ is finite, then every word $w \in L(P)$ satisfies*

$$|w| < 3r(r^2 + 1)^2.$$

*Proof.* Let $P = (pr, su, In)$ be a profile over an alphabet $\Sigma$ that defines a finite language, where $In = \{(w_1, \theta_1 t_1), \ldots, (w_m, \theta_m t_m)\}$. Moreover, assume that $k \in \{0, \ldots, m\}$ satisfies the following:

- $\theta_i \in \{<, \leq, =\}$ for every $i \in \{1, \ldots, k\}$, and
- $\theta_i \in \{>, \geq\}$ for every $i \in \{k+1, \ldots, m\}$.

Notice that if $k = 0$, then every $\theta_i$ is either $>$ or $\geq$, and if $k = m$, then every $\theta_i$ is either $<$ or $\leq$ or $=$.

If $m = 0$ or $k = 0$, then $L(P)$ is infinite, and we obtain a contradiction as we assume that $L(P)$ is finite. Hence, we have that $m \geq 1$ and $k \in \{1, \ldots, m\}$, and we define $\ell = \max\{|w_i| \mid 1 \leq i \leq k\}$. Moreover, we assume that $s$ is the length of profile $P$.

If $L(P) = \emptyset$, then the lemma trivially holds. Hence, assume that $L(P) \neq \emptyset$, and let $w$ be a word of $L(P)$ of maximum length (such a word exists since $L(P)$ is finite). For the sake of contradiction, assume that $|w| \geq 3(s^2 + 1)^2 s$. Then, considering that:

$$|pr| + |su| + \left( \sum_{i=1}^{m} |w_i|(|t_i| + 1) \right) \leq s^2,$$

$$2(\ell k + 1)^2 \cdot \left( 1 + m + \sum_{i=1}^{m} |t_i| \right) \leq 2(s^2 + 1)^2 s,$$

$$s^2 + 2(s^2 + 1)^2 s \leq 3(s^2 + 1)^2 s,$$

we conclude that:

$$|pr| + |su| + \left( \sum_{i=1}^{m} |w_i|(|t_i| + 1) \right) +$$

$$2(\ell k + 1)^2 \cdot \left( 1 + m + \sum_{i=1}^{m} |t_i| \right) \leq |w|. \quad (3)$$

Therefore, we conclude that there exists $u, w', v \in \Sigma^*$ such that:

- $w = uw'v$,
- *pr* is a prefix of $u$ and *su* is a suffix of $v$,
- $|w'| \geq 2(\ell k + 1)^2$,
- $w'$ does not have any of $w_1, \ldots, w_k$ as subword,
- the number of occurrences of $w_i$ as a subword of $u$ or $v$ is at least $t_i + 1$, for every $i \in \{k+1, \ldots, m\}$ such that $\theta_i = >$, and
- the number of occurrences of $w_i$ as a subword of $u$ or $v$ is at least $t_i$, for every $i \in \{k+1, \ldots, m\}$ such that $\theta_i = \geq$.

We know require the following "pumping lemma" for profiles (proved in the appendix)

**Lemma 5.3.** *Let $w_1, \ldots, w_m \in \Sigma^*$ with $m \geq 1$, $\ell = \max\{|w_i| \mid 1 \leq i \leq m\}$, and $w \in \Sigma^*$ such that $|w| \geq 2(\ell m + 1)^2$ and $w$ does not have any of $w_1, \ldots, w_m$ as subword. Then there exist $u, x, v \in \Sigma^*$ such that $w = uxv$, $|x| > \ell$ and the word $uxxv$ also does not have any of $w_1, \ldots, w_m$ as subword.*

By lemma 5.3, there exist $u', x, v' \in \Sigma^*$ such that $w' = u'xv'$, $|x| > \ell$ and the word $u'xxv'$ does not have any of $w_1, \ldots, w_k$ as subword. Hence, we conclude that $uu'xxv'v \in L(P)$ by the previous conditions. But then we obtain a contradiction since $|uu'xxv'v| > |w|$ (given that $w = uw'v = uu'xv'v$ and $|x| > 0$) and $w$ is an element of $L(P)$ of maximum length. $\square$

Applying Lemma 5.2 with $r = s\ell$, we conclude that every word $w \in \bigcap_{i \in I} L_i$ has length strictly less than $3s\ell((s\ell)^2 + 1)^2$. The proof of Lemma 5.2 can be found in the appendix.

*Step 4: Counting words.* The number of words over an alphabet of size $a \geq 2$ of length strictly less than $3s\ell((s\ell)^2 + 1)^2$ is bounded by $a^{3s\ell((s\ell)^2+1)^2}$. Therefore (2) holds, completing the proof of (a).

This concludes the proof of the theorem. $\square$

# 6. Results for Non-Erasing Pattern Languages

Let $\mathcal{X}$ be a countably infinite set of *variables*, and use uppercase letters $X, Y, \ldots$ to denote its elements. A *pattern* over a finite alphabet $\Sigma$, assumed disjoint from $\mathcal{X}$, is a word in $(\Sigma \cup \mathcal{X})^*$.

A *valuation* over $\Sigma$ is a mapping $\mu : \mathcal{X} \to \Sigma^*$ assigning to each variable $X \in \mathcal{X}$ a word $\mu(X) \in \Sigma^*$. The valuation is

*non-erasing* if $\mu(X)$ is a nonempty word for every $X \in \mathcal{X}$. If $p$ is a pattern and $\mu$ a valuation, we write $\mu(p)$ for the word in $\Sigma^*$ obtained by simultaneously replacing each occurrence of a variable $X$ in $p$ with the word $\mu(X)$.

Each pattern $p$ over $\Sigma$ defines a *non-erasing pattern language* over $\Sigma$ by

$$L(p) = \{\mu(p) \mid \mu \text{ is a non-erasing valuation over } \Sigma\}.$$

As shown in the next example, non-erasing pattern languages are incomparable with context-free, regular, and LTT languages.

**Example 6.1.** Consider the pattern $p = XX$. Then $L(p)$ consists of all words of the form $ww$, where $w$ is a nonempty word over $\Sigma$. This language is not context-free, and hence not regular. In turn, the regular language $a^*b^*$, which is even LTT, is not a non-erasing pattern language. □

Non-erasing pattern languages are known to have favorable properties for language identification in the limit under Gold's model (Angluin, 1980a). We show next that, despite this, finite families of such languages need not admit feasible language generation in the limit.

**Theorem 6.1.** *For every $n \in \mathbb{N}$, there exists a family $\{p_1, p_2\}$ of two infinite non-erasing pattern languages over the alphabet $\Sigma = \{a, b\}$ satisfying the following: $p_1$ is defined by a pattern of size $44n$, $p_2$ is defined by a pattern of size $2$, and $\{p_1, p_2\}$ is not $2^n$-generatable.*

*Proof.* By Proposition 2.3, it suffices to construct patterns $p_1, p_2$ such that $L(p_1)$ and $L(p_2)$ are infinite and $|L(p_1) \cap L(p_2)| = 2^n$. Consider the following patterns of size 22:

$$\begin{aligned} q &= XaXbXaabbabaXbabaabbab, \\ r &= abaabbabaXbabaabbXaXbX. \end{aligned}$$

It has been shown by Nowotka & Saarela that $|L(q) \cap L(r)| = 2$. For each $i \in [n]$, let $q_i$ be the pattern obtained from $q$ by replacing every occurrence of $X$ with a fresh variable $X_i$; define $r_i$ analogously (use the same variable $X_i$ in $r_i$ as in $q_i$). Now set

$$p_1 = q_1 q_2 \cdots q_n r_1 r_2 \cdots r_n, \qquad p_2 = YY.$$

Clearly, $L(p_1)$ and $L(p_2)$ are infinite. Moreover, since $p_2$ forces words of the form $ww$ and the two halves of $p_1$ have the same length (because $|q| = |r|$), any word in $L(p_1) \cap L(p_2)$ must be of the form

$$ww \qquad \text{with} \qquad w \in L(q_1 q_2 \cdots q_n) \cap L(r_1 r_2 \cdots r_n).$$

Thus, $ww \in L(p_1) \cap L(p_2)$ iff there exists a non-erasing valuation $\mu$ such that $\mu(q_1 q_2 \cdots q_n) = \mu(r_1 r_2 \cdots r_n)$. Since $|q_i| = |r_i|$ for each $i$, equality of the concatenations implies blockwise equality: $\mu(q_i) = \mu(r_i)$ for all $i \in [n]$.

Because $q_i$ and $r_i$ share *only* the variable $X_i$, the condition $\mu(q_i) = \mu(r_i)$ depends only on the value of $\mu(X_i)$, and the choices for different indices $i$ are independent. Furthermore, by the result of Nowotka & Saarela, for each $i$ there are exactly two non-erasing valuations for $X_i$ that satisfy $\mu(q_i) = \mu(r_i)$, i.e., $|L(q_i) \cap L(r_i)| = 2$. It follows that there are exactly $2^n$ choices for $(\mu(X_1), \ldots, \mu(X_n))$ that simultaneously satisfy $\mu(q_i) = \mu(r_i)$ for all $i$, and each such choice yields a distinct word in $L(p_1) \cap L(p_2)$ (already the $i$th block differs whenever the choice for $X_i$ differs). Hence, $|L(p_1) \cap L(p_2)| = \prod_{i=1}^{n} |L(q_i) \cap L(r_i)| = 2^n$. By Proposition 2.3, the family $\{p_1, p_2\}$ is not $2^n$-generatable.

Finally, $p_2$ has size 2, and $p_1$ has size $44n$ as it is a concatenation of $2n$ patterns of size 22. □

## 7. Conclusions and Limitations

The theorem of Kleinberg and Mullainathan was an attempt to give a completely model-free explanation of the remarkable recent success of LLMs. While conceptually appealing, this approach is subject to immediate criticism: modern architectures, such as transformers or state-space models, may not be able to simulate their generation algorithms. In our paper, we go further by showing that, regardless of the architecture choice, natural families of languages require an infeasible amount of data for generation. We identify the barrier of large finite intersections that no potential generator, including practical architectures that are used today, could surpass.

A crucial limitation of our work is that it assumes the adversarial generation of the data. However, it also leaves two natural hypotheses for an explanation of the empirical generational success of LLMs:

- training data in practice might be avoiding large intersections;

- families and grammars on average-case might avoid complex structures that we use in our lower bounds.

Which of these two alternatives is observed in the real world is a question for future empirical research.

Beyond their implications for learning theory, our results also connect to classical problems in formal language theory. In particular, the lower bounds we obtain are governed by the size of the largest finite intersection of languages in the family (see Proposition 2.3). While this quantity has received little direct attention, its analysis relies on techniques similar to those used in studying the complexity of deciding whether the intersection of a finite collection of languages is nonempty (Kozen, 1977). From this perspective, language generation in the limit offers a new lens on the complexity of language intersection.

## Acknowledgements

Kozachinskiy is supported by ANID Fondecyt Iniciación grant 11250060. Barceló and Kozachinskiy are funded by the National Center for Artificial Intelligence CENIA FB210017, Basal ANID. Arenas and Barceló are also funded by ANID Millennium Science Initiative Program Code ICN17002.

## Impact Statement

This paper presents work whose goal is to advance the field of Machine Learning. There are many potential societal consequences of our work, none which we feel must be specifically highlighted here.

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

# A. Missing proofs

# B. A pumping lemma for profiles

**Lemma B.1** (Restatement of Lemma 5.3). *Let $w_1, \ldots, w_m \in \Sigma^*$ with $m \geq 1$, $\ell = \max\{|w_i| \mid 1 \leq i \leq m\}$, and $w \in \Sigma^*$ such that $|w| \geq 2(\ell m + 1)^2$ and $w$ does not have any of $w_1, \ldots, w_m$ as subword. Then there exist $u, x, v \in \Sigma^*$ such that $w = uxv$, $|x| > \ell$ and the word $uxxv$ also does not have any of $w_1, \ldots, w_m$ as subword.*

*Proof.* Let $w \in \Sigma^*$ be a word that satisfies the hypothesis of the lemma. Moreover, let *Pref* be the set of prefixes of the words $w_1, \ldots, w_m$. Notice that $|Pref| \leq \ell m + 1$.

Assume that $w = w[1] \cdots w[n]$, where $w[i] \in \Sigma$ for every $i \in \{1, \ldots, n\}$. Then define an assignment $\tau : \{1, \ldots, n\} \to Pref$ as follows. Take position $i \in \{1, \ldots, n\}$, and let $Pref_i$ be the set of words of the form $w[i - k + 1] \cdots w[i]$ such that $k \in \mathbb{N}$ and $w[i - k + 1] \cdots w[i] \in Pref$. Observe that $Pref_i \neq \varnothing$ because $\varepsilon \in Pref_i$, and it is a finite set since $|Pref_i| \leq |Pref| \leq \ell m + 1$. Then we define $\tau(i)$ as the word in $Pref_i$ with maximum length.

Given that $|w| \geq 2(\ell m + 1)^2$ and $|Pref| \leq \ell m + 1$, there exists an element $p$ in *Pref* that is assigned to at least $2(\ell m + 1)$ positions (that is, $|\{i \in \{1, \ldots, n\} \mid \tau(i) = p\}| \geq 2(\ell m + 1)$). Let $i_0$ be the minimum position such that $\tau(i_0) = p$, and $i_1$ be the maximum position such that $\tau(i_1) = p$. Then define $x$ as the subword of $w$ between positions $i_0$ and $i_1$, including $w[i_0]$ but excluding $w[i_1]$. Moreover, define $u$ and $v$ as the remaining prefix and suffix of $w$, so that $w = uxv$. This word is depicted in the following figure, where we also show the first and last occurrences of $p$:

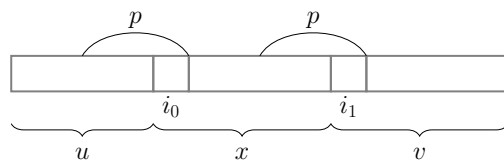

We have that $|x| > \ell$ because $x$ contains at least $2(\ell m + 1)$ positions. Hence, it remains to prove that $uxxv$ does not have any of $w_1, \ldots, w_m$ as subword. The word $uxxv$ is depicted in the following figure:

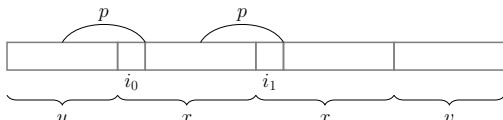

Suppose, by contradiction, that there exists $i \in \{1, \ldots, m\}$ such that $w_i$ is a subword of $uxxv$. Given that $w_i$ is not

a subword of $w$, it follows that $w_i$ is a subword of $xx$, with a nonempty part lying in the first occurrence of $x$ and a nonempty part lying in the second occurrence of $x$ (as otherwise $w_i$ would be a subword of $ux$ or $xv$, which contradicts our assumption that $w_i$ is not a subword of $w = uxv$). This gives rise to two cases for our proof, depending on how $p$ and $w_i$ are related in $uxxv$.

1. Assume first that the first symbol of $w_i$ occurs before the first symbol of $p$, as depicted in the following figure:

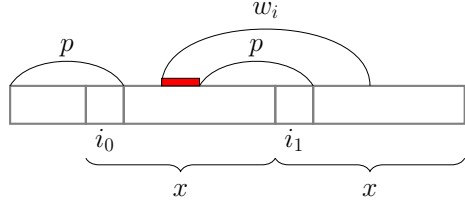

   In particular, the red region in the figure is not empty. Notice that in this case, $p$ is a subword of $w_i$, since only the last symbol of $p$ appears in the second occurrence of $x$ (at position $i_1$), and a nonempty part of $w_i$ lies in the second occurrence of $x$. Then there exists a prefix $y$ of $w_i$ such that $p$ is a proper suffix of $y$. But this leads to a contradiction since $y \in Pref_{i_1}$, $|p| < |y|$, and $\tau(i_1) = p$.

2. Assume now that the first symbol of $w_i$ does not occur before the first symbol of $p$, as depicted in the following figure:

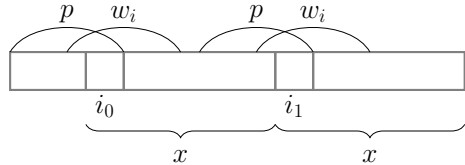

   Then given that the word $x$ occurs twice in $uxxw$, the same words appear at the beginning and at the end of the $uxxw$, as depicted in the figure. But this implies that $w_i$ is a subword of $ux$ since $|w_i| \leq \ell$ and $|x| > \ell$. We conclude that $w_i$ is a subword of $w = uxv$, which contradicts the hypothesis of the lemma.

This finishes the proof of the lemma. $\qquad\square$

