# OpenReview forum: "Language Generation in the Limit: Complexity Barriers and Implications for Learning"
_ICML.cc/2026/Conference — ICML 2026 spotlight_

### Official Review · Reviewer_pgQ5 · 2026-03-11

**Soundness:** 4
**Presentation:** 4
**Significance:** 3
**Originality:** 4
**Overall Recommendation:** 5
**Confidence:** 5

**Summary:**

The paper tries to study sample complexity in the "language generation in the limit" setting proposed by Kleinberg and Mullainatan (NeurIPS'2024 [1]), as existing results did not address feasibility of learning and only had results for identification in principle. The authors focus on uniform generation (given any ordering of inputs), i.e. the worst case sample complexity to guarantee identification given any first $m$ inputs for a fixed set of family of languages. The authors show a host of negative results across different families.
- Strictly Non-Computable for Context-Free Languages.
- Double-Exponential even for Regular Languages (irrespective of formalism used - NFA, DFA or regex)
- Single Exponential for Locally Threshold Testable Languages.
- Exponential for Non-Erasing Pattern Languages (Despite this class possessing highly favorable properties for classical grammar identification, the authors prove an exponential lower bound for its generation.)

[1] Kleinberg, Jon, and Sendhil Mullainathan. "Language generation in the limit, NeurIPS'2024.

**Compliance With Llm Reviewing Policy:**

Affirmed.

**Final Justification:**

The paper has clear merits, and my answers were clarified by the authors. The rebuttal and their answers were good enough for me to help me increase my score to a 5. I thank the authors and wish them the best.

**Key Questions For Authors:**

My questions are very related to the weaknesses I listed.

- Architectural Implications: How do these worst-case bounds interact with the inductive biases of modern architectures like Transformers or RNNs? Could the authors provide a brief discussion on what they think?
- Data Ordering: The definition of uniform generation assumes an arbitrary or adversarial data ordering. If the model instead receives examples in a structured, helpful order (e.g., a curriculum learning setup), does the sample complexity drop to something feasible? Could the authors speculate on this, if the answer to this is technically challenging to show?
- Average vs. Worst Case: The lower bounds rely on highly specific, adversarial constructions (like the Turing machine reduction or nested lexicographic blocks). Do the authors have any theoretical intuition for what the sample complexity looks like in the "average case" or for naturally occurring grammars?

**Limitations:**

The authors didn't discuss the limitations of their work. I would suggest the authors to include it themselves.
From my perspective, the main limitation is the lack of practical utility of the theory.

**Strengths And Weaknesses:**

**Strengths**

*Sample Complexity*: The paper takes a very necessary step forward in formal learning theory. While previous work showed that generation is in principle possible, asking "how much data does it actually need?" is the right question, and proving that the answer is often "an impossible amount" is an important contribution.

*Breadth*: The authors didn’t just cherry-pick one language class to make their point. They consider many families from the formal language hierarchy and/or past literature—from Context-Free Grammars down to Regular, LTT, and even Non-erasing Pattern languages.

*Correctness*: The proofs of all the statements look correct to me.

**Weaknesses**

*Still Disconnected from Practical Architectures*: While these theoretical bounds are important, practical implications are unclear. The paper proves generation is incredibly data-hungry, but it doesn't bridge this to the models we actually use. What does this mean for Transformers, State Space Models, or RNNs? This paper lacks an immediate, practical takeaway for the ML community.

*Worst-Case Analysis*: The impossibility results heavily rely on adversarial, worst-case constructions (e.g., building CFGs that simulate Turing machines, or nested lexicographic traps).  However, real-world data distributions do not have to look like these pathological edge cases. If a specific subset of CFGs doesn't exhibit these massive "intersection traps," does the sample complexity become manageable? Whether these bounds hold for the "average case" or for more naturally occurring grammars remains an open question.

*Strict Uniformity and Ordering*: The definition of uniform generation requires the learner to succeed regardless of the ordering of the examples. But in practice, the order in which a model sees data matters a lot (e.g., curriculum learning). If we relax this constraint—say, by always providing a specific, helpful ordering of examples rather than an arbitrary or adversarial one—does the sample complexity drop significantly? Current definitions of generation and uniform generation might be artificially inflating the difficulty of the task and thus the sample complexity.

---

> ### Author Rebuttal · Authors · 2026-03-30
>
> Thank you for the review!
>
> We are glad that the reviewer acknowledges the importance and breadth of our theoretical contribution. At the same time, the reviewer points out that our theoretical framework is distant from practical language generation, and indicates the necessity of refining the framework. We agree with the reviewer on this last point. In fact, proving this point was one of the goals of our paper.
>
> The theorem of Kleinberg and Mullainathan was an attempt to give a completely model-free explanation of the remarkable recent advances of LLMs. While conceptually appealing, it is subject to immediate criticism that modern architectures like transformers or state-space models might not be able to simulate their generation algorithm. In our paper, we go further – and this is related to your first question – by showing that regardless of the architecture choice, natural families of languages require a non-feasible amount of data for generation. We identify the barrier of large finite intersections that no potential generator, including practical architectures that are used today, could surpass due to a formally established limitation.
>
> However, the reviewer rightfully points out that this limitation is based on the adversarial worst-case data-generation assumption. In fact, regarding your second question, the ordering of data is not that important as we consider generators whose inputs are sets of words – that is, those that ignore the order of the input sequence and only pay attention to occurrences of words in it. Nevertheless, to obtain our lower bounds, it is crucial for the input data to come from a large finite intersection of languages from a family. What does it mean in practice in comparison to the empirical success of generative models? There are two natural hypotheses:
>
> * Training data in practice might be avoiding large intersections.
>
> * As you indicate in your last question, families and grammars on average-case might be avoiding complex structures that we use in our lower bounds.
>
> Which of these two alternatives is observed in the real word is a question for the future empirical and potentially, theoretical research (where the latter would require formalizing  a challenging notion of an ‘’average-case’’ context-free grammar, or an ‘’average-case’’ regular language). Let us also indicate a possibility that practical families could be structurally similar to locally threshold testable (LTT) languages, for which we have an upper bound – which is exponential, but at least just single-exponential, unlike regular languages and CFGs.
>
> We find that narrowing down the set of alternatives for the empirical success of generation is an important practical takeaway that would not be possible without the technical mathematical work, done in our paper. We agree with the reviewer that it is necessary to elaborate more on these matters in the limitations section.

---

> > ### Author Rebuttal · Reviewer_pgQ5 · 2026-03-31
> >
> > Thank you for your response. I have read through the other reviews, the authors responses to them and the authors response to me. Yes, I agree that the questions I have raised are probably not answerable so easily in the current scope of work. Specifically, the two hypothesis you put forth to compare your findings to empirical success also makes sense, but I guess it is not possible to go deeper than that now. I agree that this paper takes steps in important and necessary directions. I will increase my score, however I hope that the authors will add the limitations section that they promised.

---

> > > ### Author Response · Authors · 2026-04-01
> > >
> > > Thank you very much for your response. We will add a limitations section to the document as promised.

---

### Official Review · Reviewer_RBjn · 2026-03-11

**Soundness:** 2
**Presentation:** 2
**Significance:** 2
**Originality:** 2
**Overall Recommendation:** 4
**Confidence:** 1

**Summary:**

The paper revisit language generation in the limit, with an additional dimension of sample complexity: how many samples the learner needs to observe before it can perform language generation in the limit. The results vary depending on the class of formal languages. Interestingly sample complexity is bounded for languages having inherent structure.

**Compliance With Llm Reviewing Policy:**

Affirmed.

**Key Questions For Authors:**

When the original collection of languages contain both regular and context-free languages, what is the sample complexity? Is it dictated by the more difficult language?

**Strengths And Weaknesses:**

The paper is theoretically rich, and has a solid motivation: asymptotic language generation is less useful than a standard guidence of how many samples must be observed before language generation is practically valid. The paper has many theories depending on the complexity of the formal language.

One weekness I can think of is the lack of experimental evaluation. Although the focus has been theoretical, practical experiments with large language models can strengthen the results. Many of the proofs can potentially be simulated in experiments, with explicit number of samples or running time.

---

> ### Author Rebuttal · Authors · 2026-03-30
>
> Thank you for the review!
>
> As for your question, yes, the sample complexity of a family can only increase if we add more languages to the family. In that sense, sample complexity of families that include both regular and context-free languages might be uncomputable just because the same already holds for families with 2 context-free languages.
>
> We would also like to address the reviewer’s comment on experimental evaluation. We agree that an experimental evaluation is a necessary step in investigating the problem of generating a language in the limit. However, we believe that it was first necessary to connect the rich body of work on formal languages with this problem in order to understand the sample complexity of existing language specifications. In particular, our goal was to analyze how results from the study of formal languages can be applied to this setting, especially considering the most common representations and natural restrictions for language specification. Having established this theoretical foundation, which is the main contribution of this paper, we believe that a comprehensive experimental evaluation can now be developed. Such an evaluation would help understand how tight the bounds obtained in the paper are and how they can be used to identify difficult cases for LLMs.

---

> > ### Author Rebuttal · Reviewer_RBjn · 2026-04-01
> >
> > Thanks for the rebuttal. My concerns are resolved. I maintain my original score.

---

> > > ### Author Response · Authors · 2026-04-01
> > >
> > > Thank you very much for your response.

---

### Official Review · Reviewer_rt6V · 2026-03-12

**Soundness:** 4
**Presentation:** 3
**Significance:** 4
**Originality:** 4
**Overall Recommendation:** 5
**Confidence:** 3

**Summary:**

This theoretical paper characterizes the sample complexity of certain language families under the framework of language generation in the limit. This is relevant to the way LLMs are trained and used, as LLMs learn exclusively from positive examples and need only to generate (ideally novel) positive examples afterwards. The key finding is that, for multiple language families of interest, language generation has intractable sample complexity. For finite families of CFLs represented as CFGs or PDAs, the sample complexity isn't even computable. For finite families of regular languages represented as NFAs, the sample complexity is double-exponential in the size of the NFAs and the family. A similar result holds for regular languages represented as regular expressions. For finite families of locally threshold testable (LTT) languages (a subset of regular languages), the sample complexity is singly exponential. For finite families of non-erasing pattern languages, which have been shown to have favorable properties for language identification in the limit (the recognition version of language generation in the limit), sample complexity is still exponential.

**Compliance With Llm Reviewing Policy:**

Affirmed.

**Final Justification:**

I think the paper is strong in terms of soundness, originality, significance, and clarity. I weighted these with roughly equal importance. The rebuttal reinforced my positive assessment.

**Key Questions For Authors:**

1. What is the purpose of the prefixes and suffixes in LTTs? Is this simply for parity with prior definitions?
2. Can you explain the statement of Theorem 4.1 (b) in plain language?

**Limitations:**

yes

**Strengths And Weaknesses:**

Overall, an excellent and impactful paper.

Soundness: Technically solid work. I have not found issues in the proofs. The paper uses standard assumptions under the framework of language generation in the limit.

Presentation: The paper is well-written and does a good job of motivating and contextualizing the results at a high level. Some parts of the paper could be written in a more accessible way, and some of the theorem statements and proofs would benefit from some clarification in plain language.

In particular, the statement at 091 right could use some clarification; I had to think about this for quite a while.

Something that threw me off about Section 3.2 is that once you get the CFGs from the algorithm for Lemma 3.2, you can't compute the size of the intersection directly -- it might not even be finite if the TM $M$ doesn't halt. You need to infer the maximum size of the intersection, and this is what the hypothetical $m$-generatability algorithm would give you. I had to spend quite a lot of time thinking about this before it really clicked.

The statement for Theorem 4.1 (b) is extremely complex and needs to be stated in plain language somewhere.

Prop 2.3: It would be helpful to point out that $L' \in \mathcal{F}$.

Significance and Originality: This is significant and timely work; it is the first characterization of sample complexity under the framework of language generation in the limit. The proof techniques are quite clever; for example, the construction in 3.2 resembles the one used to show that checking CFG equivalence is undecidable, except they use a clever way to translate the length of the accepting computation to the size of the intersection.

---

> ### Author Rebuttal · Authors · 2026-03-30
>
> Thank you for the review!
>
> * _1. What is the purpose of the prefixes and suffixes in LTTs? Is this simply for parity with prior definitions?_
>
> It is a standard part of the definition of LTT languages that one can fix a prefix and a suffix in a profile. As such, this definition admits several equivalent logical and algebraic characterizations, see [Place et al., 2014]. We are not aware if these characterizations still hold if one drops the conditions about prefixes and suffixes. Nevertheless, let us note that our single-exponential lower bound on sample complexity (Theorem 5.1 b) uses LTT languages where prefixes and suffixes are not specified — in that sense, our lower bound is robust.
>
> * _2. Can you explain the statement of Theorem 4.1 (b) in plain language?_
>
> In Theorem 4.1, we study a question -- what is the largest possible sample complexity of a family of $m$ languages (over the binary alphabet), each recognizable by an $s$-state NFA? In item b), we give a _lower bound_, showing that for any $m,s$, this sample complexity can be as large as:
>
> $2^{(\frac{s}{k\cdot\log_2 m})^m}$
>
> where $k > 0$ is an absolute constant. That is, we construct an example of a family with $m$ languages, each recognizable by an $s$-state NFA (in fact, these NFA are deterministic in our construction) whose sample complexity is at least the quantity above.
>
> This lower bound almost matches our _upper bound_ from item a) that says that for any family of $m$ languages, each recognizable by an $s$-state NFA, its sample complexity does not exceed
>
> $2^{s^m}$

---

> > ### Author Rebuttal · Reviewer_rt6V · 2026-04-03
> >
> > Thank you for the clarifications! I will keep my (very positive) score the same.

---

> > > ### Author Response · Authors · 2026-04-04
> > >
> > > Thank you very much for your response!

---

### Decision · Program_Chairs · 2026-04-30

**Decision:**

Accept (spotlight)

**Comment:**

This theoretical paper builds upon Kleinberg and Mullainathan's paper, establishing the difference between language generation and language learning.
Along this framework, the main results of the paper concern the worst-case sample complexity for different language families, ranging from context-free grammars to locally threshold testable languages).

All reviewers and the meta-reviewer are impressed by the breadth and vision of the paper.